# Demystifying DPP III Catalyzed Peptide Hydrolysis—Computational Study of the Complete Catalytic Cycle of Human DPP III Catalyzed Tynorphin Hydrolysis

**DOI:** 10.3390/ijms23031858

**Published:** 2022-02-06

**Authors:** Antonija Tomić, Sanja Tomić

**Affiliations:** Division of Organic Chemistry and Biochemistry, Ruđer Bošković Institute, Bijenička 54, 10000 Zagreb, Croatia

**Keywords:** dipeptidyl peptides III, tynorphin, metallopeptidase activity inhibition, enzyme reaction cycle, peptide hydrolysis

## Abstract

Dipeptidyl peptides III (DPP III) is a dual-domain zinc exopeptidase that hydrolyzes peptides of varying sequence and size. Despite attempts to elucidate and narrow down the broad substrate-specificity of DPP III, there is no explanation as to why some of them, such as tynorphin (VVYPW), the truncated form of the endogenous heptapeptide spinorphin, are the slow-reacting substrates of DPP III compared to others, such as Leu-enkephalin. Using quantum molecular mechanics calculations followed by various molecular dynamics techniques, we describe for the first time the entire catalytic cycle of human DPP III, providing theoretical insight into the inhibitory mechanism of tynorphin. The chemical step of peptide bond hydrolysis and the substrate binding to the active site of the enzyme and release of the product were described for DPP III in complex with tynorphin and Leu-enkephalin and their products. We found that tynorphin is cleaved by the same reaction mechanism determined for Leu-enkephalin. More importantly, we showed that the product stabilization and regeneration of the enzyme, but not the nucleophilic attack of the catalytic water molecule and inversion at the nitrogen atom of the cleavable peptide bond, correspond to the rate-determining steps of the overall catalytic cycle of the enzyme.

## 1. Introduction

Dipeptidyl-peptidase III (DPP III; EC 3.4.14.4) is a two-domain monozinc metalloexopeptidase from the peptidase family M49 (according to the MEROPS database, http://www.merops.ac.uk accessed on 29 September 2021) that hydrolyzes dipeptides from the unsubstituted N-terminus of its substrates [1]. It is a cytosolic enzyme but has also been found in the membrane [2,3,4] and has broad specificity for peptides of varying length and composition [5,6,7,8], with tetrapeptides to octapeptides being the best substrates [5,9].

In mammalian tissues, DPP III is widely distributed and is thought to contribute to the final steps of normal intercellular protein degradation. However, the pronounced affinity for some bioactive peptides suggests more specific functions in the human organism. Cruz-Diaz et al. showed that angiotensin-(1–7) is hydrolyzed by DPP III in renal epithelial cells, where this peptide exerts a renoprotective effect [10]. A study by Pang et al. [11] showed a link between DPP III and the renin-angiotensin system (RAS) and thus a possible use of DPP III in the treatment of hypertension. In addition, there is growing experimental evidence that DPP III is involved in the development of some cancers in humans [12,13,14,15]. High levels of DPP III in the rat spinal cord, where enkephalin-synthesizing neurons are located, and its high affinity (in vitro) for opioid peptides such as enkephalins and endomorphins suggest its role in the mammalian pain regulatory system [16,17]. However, knowledge about the exact role of DPP III in these processes is not yet satisfactory.

Among natural peptides, angiotensins and enkephalins are the preferred substrates of human DPP III and bind with micromolar affinity. The experimental data, mostly their potency to inhibit hydrolysis of a synthetic substrate (e.g., Arg-Arg-2-naphthylamide or Arg-Arg-4-methoxy-2-naphthylamide) are available only for angiotensins (angiotensin II, angiotensin III, angiotensin IV, angiotensin-(1–7) and angiotensin (3–7)) [6,7,8,10,11], proctolin [18], α-melanocyte-stimulating hormone [7], dynorphin A(1–8) [16], enkephalins (Leu-enkephalin and Met-enkephalin) [6,16] and endomorphins (endomorphin-1 and endomorphin-2) [16], as well as some hemorphins (valorphin) [16] and exorphins (β-casomorphin) [16]. In addition, the µM inhibition constant was determined for Leu-enkephalin (3.65 ± 0.60 μM) [16]. 

At the same time, tynorphin (VVYPW), the truncated form of the heptapeptide spinorphin (LVVYPWT), showed an inhibitory effect on DPP III isolated from monkey brain. The inhibition of DPP III by tynorphin was predominantly competitive with a *K*_i_ value of 7.5 × 10^−8^ mol dm^−3^ [19,20]. Unlike spinorphin, which exhibits an inhibitory effect on various enkephalin-degrading enzymes (such as neutral endopeptidase; NEP, aminopeptidase; AP and angiotensin-converting enzyme; ACE), the effect on NEP, AP and ACE was found to be non-significant, suggesting that tynorphin is specific to DPP III [20]. In the same paper, Yamamoto et al. showed that tynorphin is hydrolyzed in human serum to a background level within 4 h at 37 °C. Furthermore, using single-injection ITC, Jha et al. [21] showed that tynorphin is a slow reacting substrate for recombinant human DPP III. Consequently, tynorphin should be considered a slowly-converted substrate rather than a true inhibitor. Chiba et al. examined the inhibitory activities of various synthetic hemorphin-like peptides and found that among different pentapeptides, tynorphin analogs IVYPW and WVYPW showed the strongest inhibitory activity toward recombinant DPP III with *K*_i_ values of 0.100 ± 0.011 μM and 0.126 ± 0.015 μM, respectively, wherein the *K*_i_ for tynorphin being 2.67 ± 0.58 μM [17]. They showed that these peptides non-competitively inhibit DPP III activity, and the order of *K*_i_ values as a function of amino acid in the N-terminus was Ile ≥ Trp ≥ Phe ≥ Tyr ≥ Leu ≥ Ala ≥ Val ≥ Ser ≥ Gly [17]. These findings suggest that exogenously administrated peptides such as IVYPW, WVYPW, FVYPW or YVYPW may be more efficient DPP III inhibitors than the endogenous peptides tynorphin and spinorphin.

In addition, Bezerra et al. [22] determined Δ*G* for the binding of tynorphin to DPP III in the range of −6.5 to −9.3 kcal/mol as a function of temperature (5–35 °C), with *K*_d_ in the range of 8.10 to 0.22 µM. In the same work, Leu-enkephalin was shown to bind to the inactive human DPP III variant with a *K*_d_ of 3.6 μM at 25 °C (Δ*G* of −7.4 kcal/mol).

One of the possible reasons why DPP III cleaves peptides with different sequences and sizes is its flexibility, and thus the adaptability of its binding site to accommodate molecules of different sizes. Indeed, both the experimental [22,23] and computational [24,25,26] studies revealed the existence of two different forms of DPP III, the so-called open and closed forms. MD simulations have shown the presence of various DPP III conformations in solution, with the semi-closed forms being the most abundant [25]. It was shown that the semi-closed conformers of DPP III are the most appropriate for substrate recognition, while the most compact conformation is best suited for the enzyme-catalyzed reaction [25].

Although the substrate specificity of DPP III is not well defined, a preference for (a) a positively charged N-terminus, (b) the ability (propensity) of the substrate to form beta-sheet secondary structures [24], (c) hydrophobic amino acid residues at the P1′ position [7,27] and d) a proline residue at the P1 position [16], has been observed (for the definition of peptide and enzyme subsites, see Appendix A).

Based on the DPP III substrate specificities defined above and the amino acid composition of tynorphin, one might assume that this pentapeptide would be a good substrate for DPP III. The experimental data suggest otherwise. It is known that, in addition to chemical transformation, Michaelis complex formation and/or product release can be the rate-limiting steps in the complete catalytic cycle of an enzyme, as shown for metalloproteinase-2 and aspartoacylase [28,29]. To gain a more complete insight into DPP III substrate specificity, it is very important to understand the overall catalytic cycle of the enzyme. 

Tomić et al. determined the mechanism of peptide bond hydrolysis in the active site of the human DPP III–Leu-enkephalin complex by quantum mechanical/molecular mechanical (QM/MM) modeling. Deprotonation of the water molecule by the 100% conserved E451 and nucleophilic attack of the resulting hydroxide ion on the carbon atom of the cleavage bond, followed by inversion of the amide nitrogen of the same peptide bond, were identified as the rate-determining steps with an activation energy of 14.21 kcal/mol [30]. 

On the basis of the crystallographically-determined structures, Kumar et al. [23] assumed that this so-called promoted-water mechanism is not possible in the case of the DPP III–IVYPW complex. In this complex, the carbonyl oxygen of the second peptide coordinates the metal ion, leaving no space for a water molecule. At the same time, the side chain of E451 is about 4 Å away from the cleavable peptide bond and could therefore act directly as a nucleophile, leading to the formation of an acyl enzyme-like intermediate. Such a mechanism is similar to the proposed anhydride pathway of peptide hydrolysis by the zinc-dependent enzyme carboxypeptidase A, which generates a high-energy acyl-enzyme intermediate [31]. The authors hypothesized that when an inhibitory peptide (such as tynorphin) is bound, the water molecule originally bound to the zinc ion is displaced, and the reaction follows the energetically unfavorable and, therefore, slow anhydride pathway, whereas when an efficiently cleaved peptide (such as the enkephalins) is bound, the reaction follows the energetically favorable and, therefore, faster water-mediated hydrolysis pathway.

In the present study, various molecular modeling techniques were used to determine the complete catalytic cycle for tynorphin hydrolysis in the active site of DPP III to understand its inhibitory mechanism. The reaction energy profile describing tynorphin hydrolysis was generated using the QM/MM method and compared with the previously [30], and herein, determined mechanisms of Leu-enkephalin hydrolysis, a peptide that is effectively cleaved in the active site of DPP III. Using adaptively steered molecular dynamic (ASMD) simulations, we modeled the phases of ligand binding to the enzyme active site and release of the products and obtained the corresponding potential of the mean force (PMF) profiles. Conventional MD simulations of the complexes before and after hydrolysis were performed to investigate the stability of the complex and to identify residues crucial for protein-ligand binding. Since the main purpose of this work was to understand the exact reasons why certain peptide sequences are effectively cleaved by DPP III and others are not, the steps of substrate binding and product release were also modeled for DPP III in complex with Leu-enkephalin and its product. All of this has helped us to understand the DPP III nature of inhibition by tynorphin and consequently allowed us to postulate new constraints on the substrate specificity of the enzyme. Ultimately, these results have the potential to make an important contribution to the development of new pharmacological molecules specific to DPP III that would allow us to influence its action (processes in which this enzyme is involved).

## 2. Results and Discussion

To investigate all elementary steps along the enzymatic reaction of tynorphin and Leu-enkephalin hydrolysis catalyzed by human DPP III, we combined several computational approaches in the study. Using the QM/MM methodology, we constructed a reaction profile for the DPP III catalyzed hydrolysis of tynorphin and Leu-enkephalin and compared the results with the profile previously determined for Leu-enkephalin with a slightly different starting structure. The processes of binding of tynorphin and Leu-enkephalin to the active site of the enzyme and release of their hydrolysis products into the bulky solvent were modeled by ASMD simulations, and the corresponding PMF profiles were compared to determine for which of these ligands the modeled processes are more energetically expensive.

The obtained results allowed us to postulate a new constraint on the specificity of DPP III substrates.

### 2.1. Chemical Transformation of the Substrate—QM/MM Study 

Tynorphin is tightly bound in the active site of the enzyme. Similar to Leu-enkephalin, it is antiparallel to the five-stranded β-core of the lower protein domain and interacts with the conserved H568 and Y318 via strong hydrogen bonds. Contrary to the assumption of Kumar et al. [23], tynorphin does not coordinate the catalytic zinc ion in the enzyme-substrate complex (ES), see Figure 1, but there is a water molecule that coordinates the zinc ion in this structure. Located between the positively charged metal ion and the negatively charged glutamate side chain, the water molecule is strongly polarized, and its hydrogen atom (H_w1_) is shifted towards the negatively charged carboxyl group of E451 (O_e1_-H_w1_ and O_w_-H_w1_ distances of 1.33 Å and 1.13 Å, respectively). Apparently, the p*K*_a_ value of the metal-bound water molecule is lower than that of the one in solution, as already observed in other systems [32,33,34]. It is still a major challenge to obtain the exact p*K*_a_ value of the reactive water in the active site of the enzyme [35].

To model DPP III-catalyzed tynorphin hydrolysis, we used four reaction coordinates: the distance between the second carbonyl carbon atom from the tynorphin N-terminus (C_s_) and the oxygen atom of the water molecule (O_w_) to model the first reaction step ES → TS1 → INT1, the distance between H_w1_ of the protonated E451 and N_s_ to model the second reaction step INT1 → TS2 → INT2, the H_w2_-N_s_ distance to model the third reaction step INT2 → TS3 → INT3, and, again, the H_w1_-N_s_ distance to model the last reaction step INT3 → TS4 → EP. Their values, as well as the values of several other bond lengths and dihedral angles in the obtained structures, are given in Table 1.

In the first reaction step ES → TS1 → INT1, the E451-assisted deprotonation of the water molecule is followed by the nucleophilic attack of the obtained hydroxide ion on the substrate and the change of pyramidalization at the leaving amide nitrogen, N_s_, the so-called N-inversion (Figure 1). The nucleophilic attack and the formation of the C_s_-O_w_ bond are facilitated by the strong hydrogen bonding of H568 to the O_s_ oxygen of the substrate, which results in the negative charge being pulled away from the C_s_ atom, and by the positioning of the carbonyl group for attack (i.e., the plane of the peptide bond is perpendicular to the direction of OH^−^ attack). Already in the TS1 structure, E451 is protonated, and the planarity of the peptide bond is distorted (see improper dihedral angles in Table 1). The formed tetrahedral oxyanion is stabilized by the coordination to the metal dication in the TS1 and INT1 structures (the O_s_-Zn distance of 3.00 Å in ES decreases to 2.09 Å and 2.12 Å in the TS1 and INT1 structures, respectively) and by the hydrogen bonding to H568 (O_s_-H568 distances of 1.60 Å and 1.64 Å in the TS1 and INT1 structures, respectively). Note that the N-inversion is followed simultaneously by the rotation of E451 and the formation of the hydrogen bond between the N_s_ nitrogen lone pair and the Glu451-O_e1_-H_w1_ group, as well as the reorientation of the O_w_-H_w2_ group and the formation of the O_w_-H_w2_-O_e2_ hydrogen bond (see INT1 in Figure 1). The activation energy for this step is estimated to be 12.41 kcal/mol (see Figure 2 and Appendix A).

To facilitate the cleavage of the peptide bond in the second reaction step, the proton that the activated water molecule donated to E451 in the first reaction step is transferred to the nitrogen atom of the cleavable peptide bond (N_s_). Cleavage of the C_s_-N_s_ peptide bond is achieved by gradually increasing this distance from 1.53 Å in the INT1 structure to 1.58 Å and 2.56 Å in the TS2 and INT2 structures, respectively. Simultaneously, the proton H_w2_ is transferred from the substrate molecule to the carboxyl group of E451. Once the O_w_-C_s_ bond is formed, and the configuration of the leaving nitrogen is suitable for proton transfer (i.e., the orbital of the lone pair on the N_s_ nitrogen, which should accommodate the proton, points toward E451, as observed in the INT1 structure), the system evolves continuously toward the INT2 structure without crossing an energy barrier. In the INT2 structure, the nearly planar carboxyl group (C_α_-C-O_s_-O_w_ dihedral angle of 163.5°) of the product (C-product) coordinates the zinc ion bidentately (via O_w_ and O_s_). In addition, the carboxyl group is stabilized by strong hydrogen bonds with His568 and the protonated Glu451. The amine part of the product (N-product) forms a hydrogen bond with Gly389 from the β-core of the enzyme and E451. It should be noted that the INT2 structure is only 1.78 kcal/mol less stable than the ES complex.

To complete the chemical transformation, the second water hydrogen, H_w2_, which protonates the carboxyl group of E451 in INT2, should be delivered to the amine nitrogen N_s_ of the N-product. This becomes possible by the realignment of the E451 side chain, which can be observed in the TS3 structure, and by the increase in the distance between the C_s_ and N_s_ atoms (from 2.56 Å in INT2 to 2.91 Å and 3.08 Å in TS3 and INT3, respectively). The distance between O_s_ and Zn has already increased from 2.24 Å (INT2) to 2.80 Å in TS3, indicating that the C-product coordinates the zinc ion monodentately in the last phase of the reaction. Comparing this with the ES complex, we see that the zinc coordination sphere is preserved. However, the structure of INT3 is significantly (5.26 kcal/mol) less stable than the structure of ES. The final product (EP) of tynorphin hydrolysis, whose energy is lower than those of ES and INT2 (−0.19 kcal/mol and −1.97 kcal/mol, respectively), was obtained after H_w2_ transfer to the N_s_ atom, which is accompanied by rotation of the amine group of the N-product to bring H_w2_ atom closer to O_w_, with an activation energy of 1.77 kcal/mol. In the final EP structure, the C-product monodentately coordinates the Zn^2+^ (with O_w_ atom), and the N-product, in its neutral form, is hydrogen-bonded to the oxygen atom, coordinating the metal ion (O_w_-H_w2_ distance 2.23 Å) and to the protonated E451 (N_s_-H_w1_ distance 1.58 Å). In other hydrolytic enzymes such as aspartoacylase, a very flat energy surface has been observed in the final phase of the reaction due to the fluctuation of the system between a few conformations [29]. These conformations differ in whether the C-product forms a hydrogen bond with the protonated amine group of the N-product and the catalytic glutamate is deprotonated, or whether the C-product remains separated from the N-product and the catalytic glutamate is protonated. The protonated form of the tynorphin N-product and the regeneration of the E451 side chain to a carboxylate anion, which better fit the pH conditions of a neutral solution, are most likely formed on the similar, almost flat energy surface and arise during the gradual release of the products, i.e., during their detachment from the Zn^2+^ ion.

### 2.2. Comparison with the Mechanism Determined for Leu-Enkephalin Hydrolysis 

Comparison of the DPP III catalyzed mechanisms of tynorphin and Leu-enkephalin hydrolysis shows that the same sequence of elementary steps is involved in both cases. Studies have shown that E451-assisted water addition at the carbonyl carbon atom and amide nitrogen inversion are the rate-determining steps, with activation energies estimated at 14.21 kcal/mol and 12.41 kcal/mol for the hydrolysis of Leu-enkephalin and tynorphin, respectively. Once the systems overcome this energy barrier, cleavage of the peptide bond occurs, which from an energetic point of view does not require overcoming energy barriers, and the systems relax into intermediate INT2 structure in the case of tynorphin and the product EP* in the case of Leu-enkephalin (see Figure 2 and Appendix A). If only this chemical conversion step is considered, the energy cost difference of 1.8 kcal/mol might imply that tynorphin is an even better substrate of DPP III than Leu-enkephalin. To clarify this unexpected result, we re-examined in detail each elementary step of the mechanism determined for Leu-enkephalin hydrolysis. As reported, the N-inversion step in Leu-enkephalin hydrolysis: “leads to a weakening of the hydrogen bond between the leaving amide group and G389, and to the complete loss of the hydrogen bond between the orbital of the lone pair of the leaving N_s_ atom and the water molecule in its vicinity” [30]. This water molecule treated empirically formed strong hydrogen bonds with the leaving amide group during hydrolysis. In the hydrolysis of tynorphin, no such (classically treated) water molecule was found in close proximity to the exiting amide nitrogen. In addition, MD simulations revealed a higher incidence of the water molecule near the N_s_ atom of Leu-enkephalin than of tynorphin (data not shown). This is quite expected since the amino acids at the P1 and P1′ positions of tynorphin have more branched side chains than in Leu-enkephalin and therefore more readily impede access to the water molecule.

These results motivated us to recalculate the energy profile for Leu-enkephalin hydrolysis using a slightly modified ES structure in which the water molecule, originally hydrogen-bonded to the leaving amide group, was moved away from the enzyme binding site. QM/MM calculations showed that the energy barrier for the rate-determining step in the modified system is 12.11 kcal/mol, i.e., it is 0.3 kcal/mol lower than that determined for tynorphin (see Figure 2 and Appendix A). Without the MM water molecule hydrogen-bonded to the amide group of the cleavable peptide bond in the Michaelis complex structure, Leu-enkephalin is closer to the metal ion than in the presence of this water (2.89 Å determined previously compared with 2.45 Å in this structure). This result again contradicts the assumption of Kumar et al. [23] that only peptide inhibitors coordinate the metal ion. The main geometry parameters and energies for the stationary point structures obtained in this (Figure 3) and previous work are listed in Appendix A and shown in Appendix A. We found that the steps leading from ES to TS4 imply the same set of elementary transformations as determined in our previous work (indicated by similar geometry parameters in Appendix A), while the difference was observed in subsequent steps, the crossing of the TS4 energy barrier leads to breakage of the C_s_-N_s_ peptide bond and reprotonation of E451 by H_w2_ originating from the activated water molecule. However, at the end of the last reaction step (INT3 → TS4 → EP*) determined in our previous work, the system relaxed into the structure in which C-product monodentately coordinates the zinc ion (via the O_w_ atom), and the N-product is stabilized by the hydrogen bond with G389 from the enzyme β-core and the MM water molecule near it. In the modified system in which this MM water molecule was moved away from the enzyme binding site, the system stabilized after crossing TS4 in the intermediate structure (INT4) with the C-product bidentately coordinated with zinc and N-product stabilized by a hydrogen bond with E451 and G389 (Figure 3 and Figure 4). Interestingly, the geometry of the active site is quite similar in the INT4 structure determined in reaction with the modified DPP III–Leu-enkephalin complex and in the INT2 structure determined in reaction with tynorphin (see Table 1 and Appendix A and Figure 4), but energy differences between these and the ES complexes were significantly different, 6.88 kcal/mol for Leu-enkephalin and 1.78 kcal/mol for tynorphin (Figure 2 and Appendix A). From INT4, Leu-enkephalin evolves to the final product in a manner similar to that found for tynorphin after the INT2 structure. Namely, in the final step (INT4 → TS5 → EP), a rotation of E451 and the amino group of the N-product is observed in a manner that brings the H_w2_ atom closer to the lone pair of the N_s_ atom. After TS5 is exceeded, the system relaxes into the product structure (EP), in which the C-product monodentately coordinates the zinc ion (via the O_w_ atom), and N-product is stabilized by a strong hydrogen bond with the O_s_ atom of the C-product and E451. The final product is only 0.22 kcal/mol less stable than the original ES complex. 

In summary, QM/MM calculations showed that human DPP III cleaves tynorphin according to an energetically favorable promoted water mechanism. The peptide bond hydrolysis is a two-step mechanism with INT4 being the intermediate structure in the case of Leu-enkephalin, and in the case of the tynorphin three-step mechanism with INT2 and INT3 being the intermediate structures. Since the activation energy for the first reaction step is almost two times higher than the activation energy of TS5 and TS3 in the case of Leu-enkephalin and tynorphin, respectively, and seven times higher than the activation energy of TS4 of tynorphin, we can conclude that the first reaction step is the slowest, i.e., the rate-determining step. It follows that the energy barriers of the rate-determining step of peptide bond hydrolysis for tynorphin and Leu-enkephalin differ by only 0.3 kcal/mol when the reaction starts from the ES structure without the MM water hydrogen-bonded to the leaving amide group (which is the case in ES used in our previous study). However, after cleavage of the peptide bond, an intermediate with an energy comparable to that of the ES structure (ΔΔ*E*(INT2-ES) = 1.78 kcal/mol) was obtained in the reaction with tynorphin, whereas an intermediate with energy significantly higher than the Michaelis complex (ΔΔ*E*(INT4-ES) = 6.88 kcal/mol) was obtained in the reaction with Leu-enkephalin. Interestingly, the structures of the Leu-enkephalin INT4 and tynorphin INT2 active sites are similar. The only difference is the strength of the interactions of the newly formed groups with their environment. Namely, shorter distances to E451, H568 and the zinc ion (Figure 4) indicate stronger interactions of the C-product carboxylate group and the N-product amino group with their environment as well as with each other in complex with Leu-enkephalin than with tynorphin. Based on the results of the MD simulations (see Stability of the complexes section), we hypothesized that looser binding around the cleaved peptide bond in the complex with tynorphin and better stabilization of its INT2 structure is the consequence of more favorable interactions that tynorphin side chains form with protein subsites. Monodentately coordinated C-products of tynorphin and Leu-enkephalin, with energies comparable to ES complexes (0.22 kcal/mol and −0.19 kcal/mol, respectively), were obtained after realignment of E451, rotation of the N-product amino group and hydrogen transfer exceeding the energy barriers of 6.45 kcal/mol and 1.77 kcal/mol in the case of tynorphin, and 5.66 kcal/mol in the case of Leu-enkephalin. Overcoming a higher energy barrier (6.45 kcal/mol and 5.66 kcal/mol) involves breaking the Zn-O_s_ coordination bond, whereas the energy barrier of 1.77 kcal/mol involves further reorganization of the active site to better stabilize the newly formed N-product. In both complexes, the amine portion of the product is stabilized by hydrogen bonding with the C-product in addition to hydrogen bonding with the active site of the enzyme and E451. It is important to note that while the activation energies for the steps following the cleavage of the peptide bond and leading to the formation of the final product do not differ significantly in the studied reactions, this is not the case for the reverse reactions. Namely, the reverse reaction that involves crossing TS5 from EP to INT4 (Δ*E*^#^ = 12.32 kcal/mol) in complex with Leu-enkephalin is more difficult to appear in the enzyme active site than the reverse reaction that involves crossing TS4 from EP to INT3 (Δ*E*^#^ = 7.22 kcal/mol) and TS3 from INT3 to INT2 (Δ*E*^#^ = 2.97 kcal/mol) in complex with tynorphin. Rather low energy barriers (compared to rate-determining step) separating tynorphin EP and INT2 structures suggest that the system can easily move from one state to another. Since INT2 represents the intermediate state in the tynorphin hydrolysis that leads to the formation of a final product that leaves the enzyme active site and makes room for a new substrate, we hypothesized that the reason tynorphin is a poor DPP III substrate is because of the substantial stabilization of the intermediate structure by the bidentate coordinated C-product, which slows the formation of the final products that can leave the active site of the enzyme. However, to obtain a complete picture, we also studied substrate binding and the kinetics of product release. 

### 2.3. Stability of the Complexes 

The equilibrated structures of the protein-substrate and protein-product complexes were simulated for 100 ns using conventional MD simulations. The stability of the complexes was evaluated by analyzing the geometric values (RMDS and RMSF) and by calculating the MM/PBSA energies as an approximation to the binding free energy.

No major changes in the enzyme structure or in the position of the ligands within the DPP III active site were observed during the MD simulations (RMSD profiles in Appendix A). The enzyme remained in its compact form (Appendix A), and the P1′ and P2 residues of the substrates and C-products bound antiparallel to the five-stranded β-sheet of the lower protein domain (Appendix A), with the N-products being slightly more flexible than the substrate molecules and the C-products (see RMSF profiles in Appendix A).

The MM/PBSA energies (see Table 2) show a slightly higher affinity of DPP III for tynorphin (−26.56 kcal/mol) than for Leu-enkephalin (−17.73 kcal/mol), which agrees with preliminary ITC measurements (data not yet published). Moreover, the stability of DPP III with the products of tynorphin hydrolysis (−27.88 and −34.40 kcal/mol for N- and C-products, respectively) is significantly higher than with the products of Leu-enkephalin hydrolysis (−6.97 and −18.13 kcal/mol for N- and C-products, respectively).

To elucidate the differences in total binding energies and identify residues critical for ligand binding, a per-residue energy decomposition was performed using the MM/GBSA approach (Appendix A). The major energy contributions of amino acids in protein to substrate and product affinity are shown in Figure 5. It is clear that stabilization of both substrates and products is achieved by the same group of enzyme residues.

The highest stabilization of the ligands was achieved by interactions of their charged N- and C-termini with the highly conserved E316 and R669, respectively. R669, part of the S3′ subsite of the protein, interacts with the P3′ residue of the peptide via strong hydrogen bonds with its C-terminal carboxylate group, and in the case of the complex with tynorphin, the guanidine group additionally forms a cation-π interaction with the indole ring of Trp (P3′). Although quantitative modeling of cation-π interactions is challenging for additive MM force field methods, several studies have shown that they can adequately capture the structures of cation-π adducts [36,37]. The side chains of P3′ residues of Leu-enkephalin and tynorphin species are oriented in opposite directions of the interdomain clef throughout the simulations of the complexes. That is, leucine interacts with the F443 residue of the upper protein domain, whereas tryptophan interacts with the R669 residue of the lower protein domain. This geometry is consistent with geometries found in available X-ray complex structures. Indeed, the X-ray structures of pentapeptides bound to the active site of DPP III show that the side chain of P3′-amino acid residue is bound by amino acid residues from the upper protein domain in complexes with endomorphin-2 (pdb code: 5EHH), Leu-enkephalin (pdb code: 5E3A) and Met-enkephalin (pdb code: 5E33), while in all complexes with tynorphin (pdb code: 3T6B and 3T6J) or its derivatives (pdb codes: 5E3C and 7OUP) tryptophan interacts with R669 from the lower protein domain.

Further on, a notable difference in per-residue binding energies in the protein-substrate complexes is observed for F109, E316, Y318, G389, H450, H455 and E512. While E316, G389, H450 and E512 stabilize tynorphin more strongly, F109, Y318, H455 and H568 interact more strongly with Leu-enkephalin. E316 is part of the triad of protein S2 subsite (E316-N391-N394) and anchors the N-terminus of the substrate (P2 residue) in the enzyme binding cleft through strong hydrogen bonds, while G389, H450 and E512 form the protein S1′ subsite. The G389 backbone is hydrogen-bonded to the amino acid backbone at the P1′ position and is involved in the antiparallel binding of the peptide. The side chain of the P1′ tyrosine residue of tynorphin forms a hydrogen bond with E512 and stacking interactions with H450, neither interaction being possible in the case of the P1′ glycine residue of Leu-enkephalin. On the other hand, the side chains of the Leu-enkephalin residues P2-tyrosine and P2′-phenylalanine undergo stacking interactions with H455 and F109, respectively, while the tynorphin residue P2-valine CH-π interacts with H455. Both Y318 and H568, which favor Leu-enkephalin over tynorphin, are involved in ligand stabilization of the P1 residue, and their importance was recognized by QM/MM calculations.

The distribution of the per-residue binding energy contributions determined in the complexes with the hydrolysis products is similar to that for the complexes with the substrate. In the case of the N-products, A388 and S408 contribute more strongly to the binding energy in the complex with the tynorphin products and F443 and N394 in the complexes with the Leu-enkephalin products, while the difference in the contributions of E512 to their energies is insignificant. In contrast to the enzyme-substrate complexes, a significant stabilization of the N-products by the E451 carboxyl group is observed, which is more pronounced in the complex with the tynorphin product. As for the differences in the stabilization of the C-products, the contribution of Y318, which is higher for the tynorphin C-product, is the most significant.

We gained further insight into the differences in substrate binding by looking at the interaction of their amino acid residues with the DPP III (Figure 6). Thus, a more favorable energy contribution to the free energy of binding of the side chains is observed for the P2 and P2′ residues in complexes with Leu-enkephalin species and the P1′ and P3′ residues in complex with tynorphin species (see Figure 6). This is mainly due to the interactions between the aromatic amino acid residues of the ligands and the S2, S1′, S2′ and S3′ protein subsites. After hydrolysis, a significant change in the stabilization of the P1 and P1′ residues is observed. After the formation of the carboxylate anion at the P1 position, the contribution of these residues to the binding energy increased dramatically, mainly due to the strong electrostatic stabilization by the positively charged zinc ion. In the case of the tynorphin molecule, an additional contribution comes from the van der Waals interactions of the valine side chain and the S1 subsite.

While the substrates did not coordinate the zinc ion during the simulations, the carboxylate group of the C-products coordinated the zinc ion mostly bidentately in the complex DPP III–tynorphin products and monodentately in the complex DPP III–Leu-enkephalin products (see Appendix A). This is consistent with the QM/MM results, which indicated two minima during tynorphin hydrolysis with energy comparable to the ES system (INT2 and EP), with the carboxylate portion of the product coordinating the zinc ion bidentately in the INT2 system and monodentately in the EP system. In the Leu-enkephalin hydrolysis reaction, only the minimum with the C-product coordinating the zinc ion monodentately was observed in this and the previous study [30]. The formation of the protonated amino group at the S1′ subsite destabilized the binding of the P1′ residue (Figure 6). This is mainly due to an unfavorable contribution of solvation to the binding free energy. The effect is more pronounced for Leu-enkephalin than for the tynorphin N-product, where the favorable stabilization of the P1′ tyrosine side chain by the S1′ subsite residue partially offsets the unfavorable desolvation. 

In summary, the results of conventional MD simulation indicate slightly better stabilization of tynorphin than Leu-enkephalin in the enzyme active site. This difference is even more pronounced for complexes with the products of hydrolysis. Per-residue binding free energies revealed that the aromatic amino acids at the P2 and P2′ sites of Leu-enkephalin and at the P1′ and P3′ sites of tynorphin are better stabilized by the protein S2 and S2′ or S1′ and S3′ subsites than their aliphatic counterparts. Moreover, the MM/GBSA energies show that the C-product of tynorphin is better stabilized in the enzyme binding site than that of Leu-enkephalin because its P1 residue interacts more favorably with the enzyme, not only through its bidentate coordination with the zinc ion, but also through the interactions that the valine side chain makes with the protein environment. This is also true for the N-product, as tyrosine and tryptophan, the P1′ and P3′ residues of tynorphin are better stabilized with the protein environment compared to glycine and leucine, the P1′ and P3′ residues of Leu-enkephalin.

### 2.4. Substrate Binding and Product Release Steps

ASMD simulations were used to investigate in detail possible pathways of ligand unbinding and their energetics, and the relevant binding and unbinding mechanisms were elucidated by analyzing the unbinding pathways.

The choice of the reaction coordinate used in the ASMD simulation (see SI) was based on the position of the tunnels (channels) determined by the software CAVER 3.0 [38] (see Appendix A), which identified three main channels. One is located near the S2 subsite of the protein (where the N-terminus of the peptide is bound), with the entrance to the protein aligned with amino acid residues from both protein domains: Tyr196, Ser317, Gly323, Ser324, Asn394, Asp496 and Ser497 (tunnel 1). The other two tunnels have entrances to the protein located on the opposite side of the interdomain cleft, near the so-called hinge region. The entrance to one of them is located between the unstructured part of the hinge region (Ala416-Gln420) and the adjacent loop Lys666-Lys670 (tunnel 2), while the entrance of the other is located between the aforementioned loop (Lys666-Lys670) and the unstructured part of the protein N-terminus (tunnel 3). We have assumed that the C-products and the substrates can use tunnel 1 to exit the enzyme. For the C-product, this is the closest channel to exit the protein, as the N-product spatially blocks the other two. As mentioned earlier, the MD simulations have indicated the presence of various DPP III conformations in solution, with the semi-closed forms being the most abundant and best suited for substrate recognition [25]. Therefore, we have assumed that the substrate enters the enzyme binding cleft of the semi-closed protein by choosing a similar tunnel whose entrance is in the more distant part of the interdomain cleft. If the C-product is bound in the enzyme binding site, the N-product would likely leave the enzyme at the opposite site, either through tunnel 2 or tunnel 3.

In order to successfully track the release of substrates and their products from the enzyme binding site, various reaction coordinates were used (see SI for details).

The profiles of the calculated PMFs (Figure 7) indicate that the release of ligands, especially substrates, from the binding site of the enzyme into the external solvent involves a rather high energy cost. On their way through the protein to be released, the ligands are attracted and repelled by the surrounding residues so that their conformation, as well as the appearance of the tunnel surface, is constantly changing. In the following discussion, we have focused on the relative differences in the free energy curves describing ligand translocation rather than their absolute values.

The free energy curves and the corresponding barriers for the substrate unbinding process indicate a slightly more favorable binding of tynorphin compared to Leu-enkephalin in the active site of the enzyme (the calculated PMF are 168.15 and 164.96 kcal/mol, respectively). This is consistent with the MM/PBSA binding free energy calculations (Table 2). While Leu-enkephalin left the enzyme binding site without significant opening of the protein, the release of tynorphin was accompanied by a remarkable decrease in protein globularity (see Rg profile in Figure 7). The first step in the process of substrate release was the disruption of the hydrogen bond network holding the peptide antiparallel to the β-strand of the lower protein domain. After detachment, Leu-enkephalin left the enzyme binding site in what could be described as a straight-line movement through the entrance of tunnel 1, whereas tynorphin first slid deeper into the interdomain cleft, followed by the further opening of the protein with its C-terminus tightly bound to the protein S3′ subsite (see Appendix A). Finally, tynorphin exited the binding site of the enzyme through tunnel 3. Cation-π interactions between the tryptophan indole group of P3′ and the guanidine group of R669, as well as hydrogen bonds between R669 and the tryptophan carboxylate group of P3′, appear to anchor the C-terminus of the peptide in the hydrophobic pocket of the protein and prevent the exit of the substrate (Appendix A). These results suggest significant stabilization of the tynorphin P3′ residue (Trp) with the S3′ subsite of the protein, which was not observed in ASMD simulations of Leu-enkephalin release. 

ASMD simulations showed that both the C- and N-products leave the protein binding site without significant protein opening, and the corresponding PMF profiles showed that both products of each ligand bind to the enzyme binding site with similar strength. This allowed us to hypothesize that the products (C- or N-) can leave the binding site either simultaneously or sequentially in any order. However, the energy required to pull out each of the tynorphin products is about 15 kcal/mol higher than the energy required for the Leu-enkephalin products, implying that the tynorphin products are better stabilized in the enzyme binding site than the Leu-enkephalin products. This is also consistent with the calculations of MM/PBSA energies (Table 2).

The PMF increases more steeply than that of the N-product in the initial phase of the release of C-products, both Leu-enkephalin and tynorphin, indicating their strong interactions with active site residues. Indeed, in the initial phase of the release of the C-product, the hydrogen bonding network connecting the C-product to the β-strand of the lower protein domain and the coordination bond with the zinc ion must be broken. After that, both Leu-enkephalin and tynorphin, C-products left the enzyme binding site more or less spontaneously via tunnel 1 (indicated by black lines in the case of Leu-enkephalin and blue lines in the case of tynorphin in Figure 7 and structures in the top and bottom right of Appendix A). Interestingly, a similar PMF profile was obtained when the tynorphin C-product exited the enzyme binding site via the entrance adjacent to tunnel 1 (black line in Figure 7 and structure in the bottom left of Appendix A). It can be seen that the cleavage of the tynorphin C-product is a stepwise process, i.e., as it leaves the binding site of the enzyme, its coordination changes from bi-coordinated to mono-coordinated to complete cleavage (Figure 7, bottom right). In contrast to the Leu-enkephalin C-product, which leaves metal ion coordination after 1/5 of the ASMD simulation by breaking a single coordination bond, the tynorphin C-product coordinates zinc during the 1/2 of the ASMD simulation. Altogether, these results suggest that the rate-determining step in the release of the C-product corresponds to its detachment from the lower domain β-sheet and the zinc ion.

As for the release of the N-products, a monotonic increase in the PMF profiles was observed until the point where they completely leave the enzyme. From the free energy curves and the corresponding barriers, it can be concluded that the N-product of tynorphin is more tightly bound to DPP III than that of Leu-enkephalin. While the Leu-enkephalin N-product leaves the interior of the enzyme via tunnel 2, this does not seem to be possible for the tynorphin N-product since its P3′-tryptophan residue interacts strongly with the highly conserved R669, which closes this tunnel (Appendix A). Instead, the tynorphin N-product exits the enzyme either via tunnel 3 (red curve in Figure 7) or via the entrance formed directly above the hinge residues, i.e., between the protein hinge and the upper domain (orange curve in Figure 7), energetically more demanding pathways. Detailed examination of the two ASMD simulations monitoring the release of the tynorphin N-product revealed that the tryptophan residue P3′ forms stable hydrogen bonds and cation-π interactions with the guanidino group of R669 almost throughout the simulation (Appendix A), which appears to prevent the ligand from exiting via tunnel 2. 

In summary, the ASMD simulations indicate a slightly more favorable binding of tynorphin compared to Leu-enkephalin in the active site of the DPP III. However, the difference of ~15 kcal/mol in the energy required to displace the C- and N- products of these substrates from the active site into the bulky water suggests that the regeneration step of the enzyme is the rate-determining step in product release. Namely, the tynorphin products are better stabilized in the active site, leading to a slower regeneration of the enzyme and allowing the entry of a new substrate. 

## 3. Computational Methods

### 3.1. QM/MM Calculations

Reaction profiles were determined for human DPP III in complexes with two opioid peptides, tynorphin and Leu-enkephalin.

The optimized structure of the closed DPP III in complex with the peptide tynorphin (Val-Val-Tyr-Pro-Trp) obtained after 10 ns of MD simulation (as described in our previous publication) [39] was used as a template for building the initial structure for QM/MM calculations of the hydrated DPP III–tynorphin system. Namely, from the equilibrated solute-solvent structure, only 3454 water molecules forming the first and second solvation spheres of the enzyme were retained, while the positions of Tyr318, Glu451 and His568, as well as those of the zinc-coordinating residues and tynorphin, were adjusted using the complex structure determined by X-ray diffraction (pdb code: 3T6B) as a template. For this purpose, the backbones of the simulated and experimental structures were aligned. The water molecule coordinated to the zinc ion (directly involved in the chemical reaction) was added manually, according to its position in the previously studied ES structure of the DPP III–Leu-enkephalin complex [30].

The QM/MM optimized structure of the hydrated DPP III–Leu-enkephalin complex determined in our earlier work [30] was used as the initial structure for QM/MM calculations of the same system performed in this work with one difference. The “MM treated” water molecule, which stabilized the amide group of the cleavable peptide bond in the original system, was moved away from the active site of the enzyme in the system studied here. For details of the QM/MM optimization protocol, see SI.

### 3.2. Adaptive Steered MD Simulations

MD simulations are useful for investigating possible pathways for the binding and release of ligands to macromolecules. In ASMD, an external force is applied to the selected reaction coordinate to allow the system to transform in the desired direction during MD simulations [40,41,42,43,44]. The average non-equilibrium work applied to the system during ASMD is called the potential of mean force (PMF) and, according to the Jarzynski equality, reflects the relative free energy of binding (Δ*G*) of the ligand [45].

Starting points in our ASMD simulations were QM/MM optimized structures of the Michaelis complex (ES) and the enzyme-product complex (EP) obtained from the Leu-enkephalin and tynorphin hydrolysis study. The binding of the substrate (Leu-enkephalin and tynorphin) to the enzyme active site and the release of the products were modeled by tracking the ligand exit using ASMD simulations. For the latter, we separately simulated the release of the carboxylate part of the product (C-product) and the amino part of the product (N-product), but in the presence of the other product (N- or C-product) in the binding site.

All simulations were carried out using the AMBER20 MD package [46] and the ff14sb force field [47]. For the zinc ion, the extended 4-ligand hybrid parameters for bonded/non-bonded compounds derived in our previous work were used [48]. A completely solvated system was minimized, followed by heating, density equilibration and productive MD simulations. For a detailed description of the preparation of the systems and the calculation procedures used, see SI. 

### 3.3. Conventional MD Simulations

The equilibrated structures used as starting structures for ASMD simulations with substrate or product bound in the enzyme binding site were also subjected to conventional simulations at constant pressure and 300 K (*NpT* ensamble) for 100 ns. The other conditions were the same as in the last phase of equilibration.

All (conventional and adaptive steered) MD simulations were performed using the CUDA-enabled graphic processing units (GPUs) version of pmemd in the AMBER20 program package [49,50]. 

### 3.4. MM/P(G)BSA Calculations

The free energies of binding of the substrates and products were calculated using the MM/PBSA (Molecular Mechanics Poisson-Boltzmann Surface Area) [51] approach implemented in the AMBER20 program. MM/PBSA calculations were performed for all simulated systems (DPP III in complex with two substrates and its hydrolytic products) using a single trajectory approximation. Calculations were performed for the enzyme with dielectric constant 2.0 immersed in the solvent with dielectric constant 80. The ion concentration was 0.1 M. The polar component of the enthalpy of solvation was calculated by the Poisson-Boltzmann method, and the nonpolar component was determined by Δ*H*_nonpol_ = γSASA + β, where the surface area accessible by the solvent (SASA) was calculated using the MolSurf program [52]. The surface tension γ and the offset β were set to the standard values of 0.0378 kcal (mol Å^2^)^−1^ and 0.5692 kcal mol^−1^, respectively. The zinc charge was +1.21 e. 

To elucidate the differences in binding energies and identify the residues critical for ligand binding, the free energy of binding between the protein and the ligand was decomposed into the contribution of each residue using the MM/GBSA (Molecular Mechanical Generalized Born Surface Area) approach implemented in the AMBER20 software. Residue-based MM/GBSA free energy analysis was performed to highlight the residues relevant to the stabilization of the ligand (before and after the reaction). Calculations were performed for the enzyme with a relative permittivity of 1.0 immersed in a solvent with a relative permittivity of 80.0.

Both MM/GBSA and MM/PBSA calculations were performed for the last 40 ns of the MD simulations (from the 60th to the 100th ns) under *NpT* conditions, sampling the structures every 20 ps. We calculated the contribution of enthalpy to binding free energy, since conformational entropy is usually neglected due to its high computational cost when only the relative binding free energies of similar ligands are needed [53].

## 4. Conclusions

To achieve the goal of this study and demystify the DPP III catalyzed hydrolysis of tynorphin, we applied quantum molecular mechanics calculations followed by classical and adaptive steered molecular dynamics simulations of the complexes of human DPP III with tynorphin and Leu-enkephalin and their products and examined the stages of substrate binding to the enzyme active site, chemical conversion and release of the final product. We found that tynorphin is cleaved in the active site of DPP III by the same reaction mechanism determined for Leu-enkephalin, with a similar activation energy determined for the rate-determining step. However, the energies of the structurally very similar intermediate structures obtained after cleavage of the peptide bond, INT2 in the case of tynorphin and INT4 in the case of Leu-enkephalin, are quite different. While the INT2 structure of tynorphin has energy similar to the initial, ES and final EP structures, the Leu-enkephalin INT4 structure is almost 7 kcal/mol less stable than structures of the corresponding ES and EP complexes. The intermediates INT2 and INT3 in the reaction of tynorphin hydrolysis can be considered as hydrolysis products in which the C-product coordinates the zinc ion in a bidentate manner. Therefore, the active site DPP III can stabilize different product structures of tynorphin, in which the C-product coordinates the zinc ion either bidentately or monodentately. As shown by the residue-based MM/GBSA calculations, tynorphin side chains, especially those of residues P1 and P1′, interact more strongly with the protein subsites than Leu-enkephalin. Since, in both complexes, the activation energies of the forward reaction leading from the INT2 structure of tynorphin and the INT4 structure of Leu-enkephalin to the product state (EP) are almost the same and products with similar energies to the Michaelis complexes are obtained after hydrolysis, we can conclude that one of the reasons why tynorphin is a poor DPP III substrate is because of the considerable stabilization of its INT2 structure. Alternatively, we could say that the high-energy Leu-enkephalin intermediate INT4 (which is only slightly lower than the preceding transition state) promotes the forward reaction. The destabilization of intermediates has already been observed and accepted as one of the principles of enzyme catalysis [54,55]. Indeed, during the hydrolysis of tynorphin, the formation of the end products that can leave the active site of the enzyme to make room for a new substrate is slowed down because the relatively low energy barriers between the EP and INT2 structures allow the system to easily switch from one state to the other. On the other hand, in the reaction with Leu-enkephalin, the activation energy of the forward reaction (INT4 → TS5 → EP) is two times lower than that of the reverse reaction, which makes the reverse reaction less likely. It appears that lowering the energy of the reaction intermediate slows down the overall reaction.

The second reason why tynorphin is a poor DPP III substrate is the strong stabilization of the hydrolysis products in the active site of the enzyme, which is much stronger than in the case of Leu-enkephalin. Both ASMD simulations and MM/PBSA calculations showed that the tynorphin C- and N-products bind more tightly into the active site of the enzyme than the Leu-enkephalin products. According to MM/GBSA calculations, the difference in product stabilization was mainly the result of a better stabilization of the P1 and P1′ residues of the tynorphin products by the enzyme than of the Leu-enkephalin products. Unlike the Leu-enkephalin C-product, which coordinates the zinc ion monodentately, the tynorphin C-product coordinates the zinc ion bidentately (the bidentate coordinated INT2 was also confirmed by QM/MM calculation), and its P1-valine side chain interacts with the S1 protein subsite. The aromatic side chain of P1′-tyrosine residue of the tynorphin N-product is also better stabilized with the protein environment than glycine, the P1′ residue of Leu-enkephalin. Consequently, we have found that the release of the products to the outside of the protein is more energetically demanding in the case of tynorphin than in the case of Leu-enkephalin.

The overall conclusion of the study as to why tynorphin is a poor substrate compared to Leu-enkephalin is not the difference in the reaction mechanism, but the significantly higher stabilization of the products of tynorphin hydrolysis, which impede the regeneration of the enzyme. That is, the energies required for nucleophilic attack of the catalytic water molecule and inversion at the nitrogen atom of the cleavable peptide bond, which corresponds to the rate-determining step of peptide hydrolysis by DPP III, are similar for both ligands. Until now, the mechanism of inhibition of DPP III hydrolysis by tynorphin was poorly understood. Therefore, our results have the potential to make an important contribution to drug design and development.

## Figures and Tables

**Figure 1 ijms-23-01858-f001:**
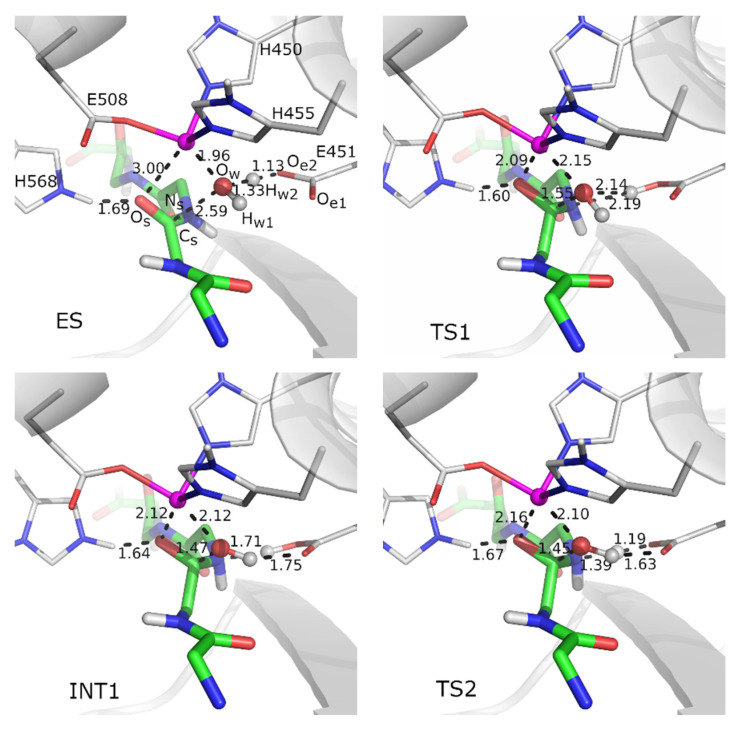
Stationary points determined for peptide bond cleavage in the hydrated DPP III–tynorphin complex. Optimized structures of the enzyme-substrate complex (ES), transition states (TS1–TS4), intermediates (INT1–INT3) and product (EP) obtained by B97D/[6-31G(d) + LanL2DZ-ECP] calculations for the E451-assisted peptide bond hydrolysis in human DPP III. The side chains of amino acid residues H450, E451, H455, E508 and H568, and the substrate (main chain only), which are part of the QM region, are shown as sticks with carbon atoms colored grey and green, respectively. The zinc ion is shown as a magenta sphere, while the water molecule QM is treated as a ball and stick representation. Coordination bonds are shown with magenta rods, and the distances between the selected pairs of atoms are given in angstroms. Only polar hydrogen atoms are shown.

**Figure 2 ijms-23-01858-f002:**
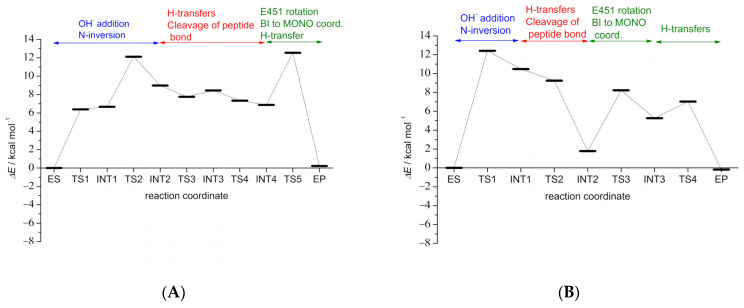
Energy profile for the hydrolysis of Leu-enkephalin (**A**) (there is no MM water molecule hydrogen bound to the amide nitrogen atom of the cleavable peptide bond) and tynorphin (**B**) in the active site of human DPP III. Calculations were performed at B97D/[6-31G(d)+LanL2DZ-ECP] + ZPVE_B97D/[6-31G(d)+LanL2DZ-ECP]_ level of theory.

**Figure 3 ijms-23-01858-f003:**
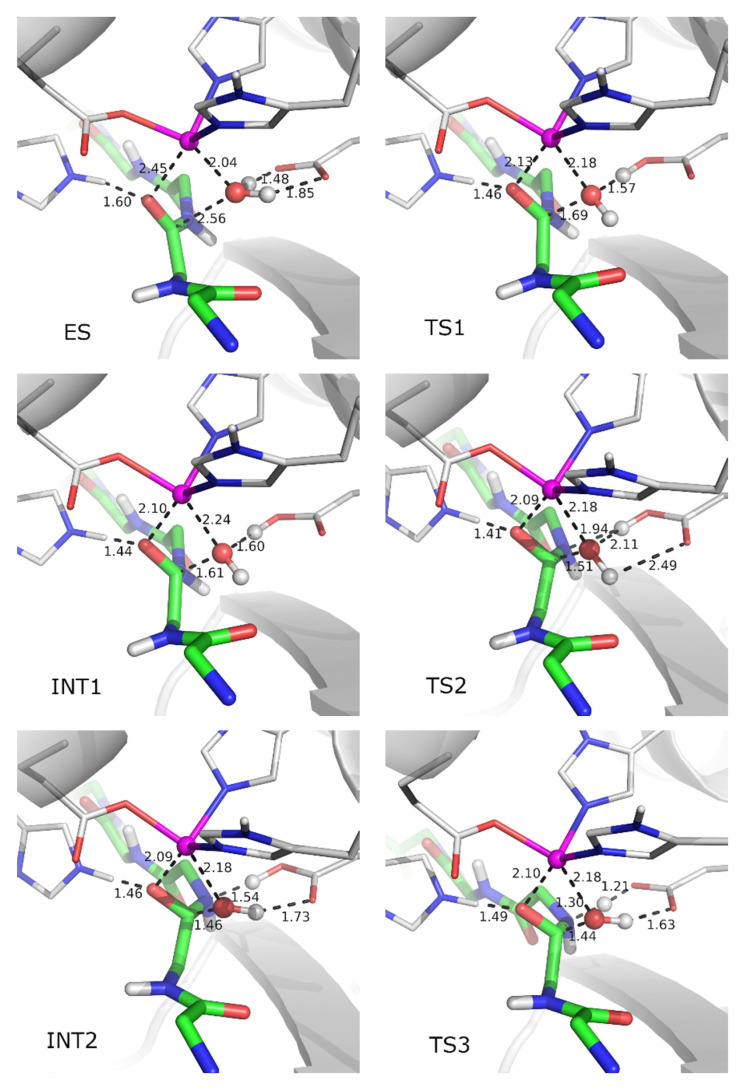
Stationary points determined for peptide bond cleavage in the hydrated DPP III–Leu-enkephalin complex. In this structure, the water molecule, originally hydrogen-bonded to the leaving amide group, was moved away from the enzyme binding site. The optimized structures of the enzyme-substrate complex (ES), transition states (TS1–TS5), intermediates (INT1–INT4) and product (EP) are shown. The side chains of amino acid residues H450, E451, H455, E508 and H568, and the substrate (main chain only), which are part of the QM region, are shown as sticks with carbon atoms colored grey and green, respectively. The zinc ion is shown as a magenta sphere, while QM treated water molecule in ball and stick representation. Coordination bonds are shown as thick magenta rods and the distances of the selected atom pairs are given in angstroms. The names of the atoms correspond to those in Figure 1.

**Figure 4 ijms-23-01858-f004:**
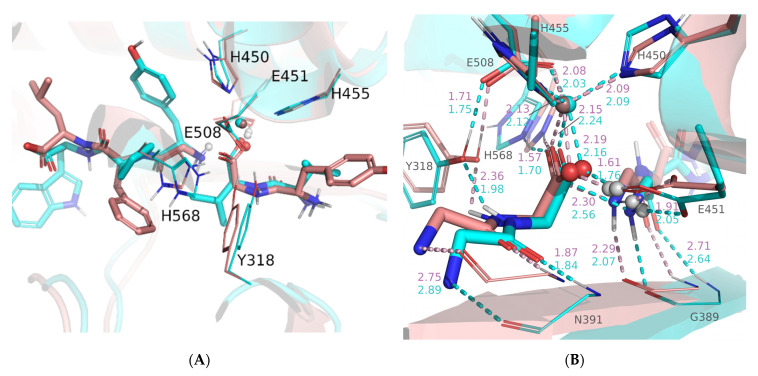
Overlay of INT4 (pink) and INT2 (cyan) structures obtained from QM/MM calculated reaction mechanisms of Leu-enkephalin (modified system in which the water molecule originally hydrogen-bonded to the leaving amide group was moved away from the enzyme binding site) and tynorphin hydrolysis by human DPP III, respectively. The alignment is based on Cα atoms of H450, E451, H455, E505 and H568. (**A**), active site of the enzyme with substrate (sticks) and quantum mechanically-treated amino acids (lines, side chains only). (**B**), magnified view of the quantum mechanically-treated amino acid residues (Y318, H450, E451, H455, E505 and H568; side chains only) and substrate (backbone atoms only), shown as thinner and thicker sticks, respectively, and backbones of amino acids G389 and N391, which form the β-core of the protein, shown as lines. The values of the selected distances (dashed line) are indicated. The zinc ion is shown as a sphere. Only polar hydrogen atoms are shown.

**Figure 5 ijms-23-01858-f005:**
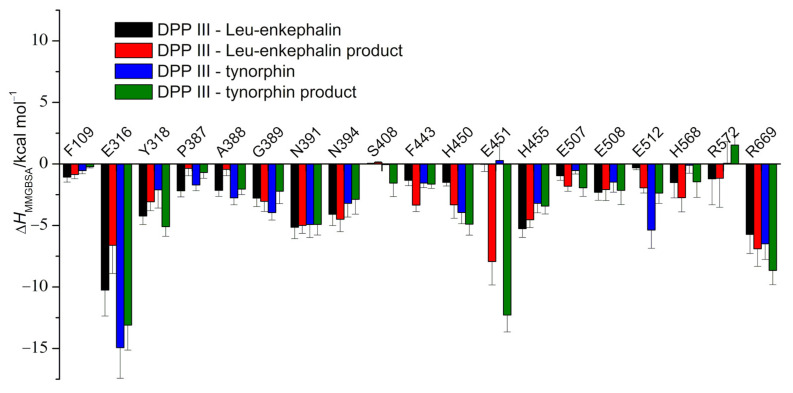
MM/GBSA per-residue binding free energies of DPP III in complex with various ligands (substrates and products of their hydrolysis) calculated on 40 ns long trajectories obtained from conventional MD simulations (60th to 100th ns). Values are given for residues with energy contributions above ±2 kcal/mol.

**Figure 6 ijms-23-01858-f006:**
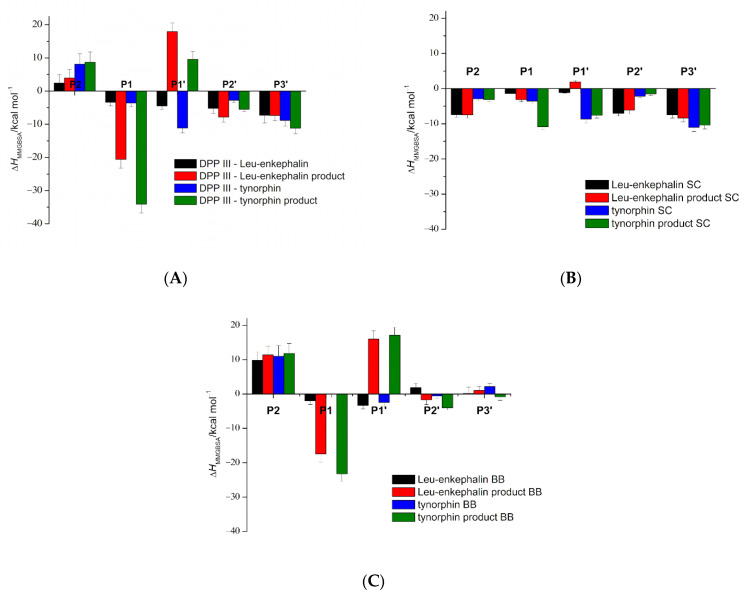
(**A**), per-residue binding-free energies of the bound peptides with entire protein calculated by MM/GBSA on 40 ns long trajectories (60th to 100th ns) obtained from conventional MD simulations of the complexes (with peptides and their hydrolytic products). The amino acid residues of the peptide ligand are designated as P1 to Pn and P1′ to Pn′, counting from the scissile peptide bond to the N- and C-termini of the peptides, respectively. The contribution of each residue side chain (SC; (**B**)) and backbone (BB; (**C**)) to the total per-residue binding energy is shown.

**Figure 7 ijms-23-01858-f007:**
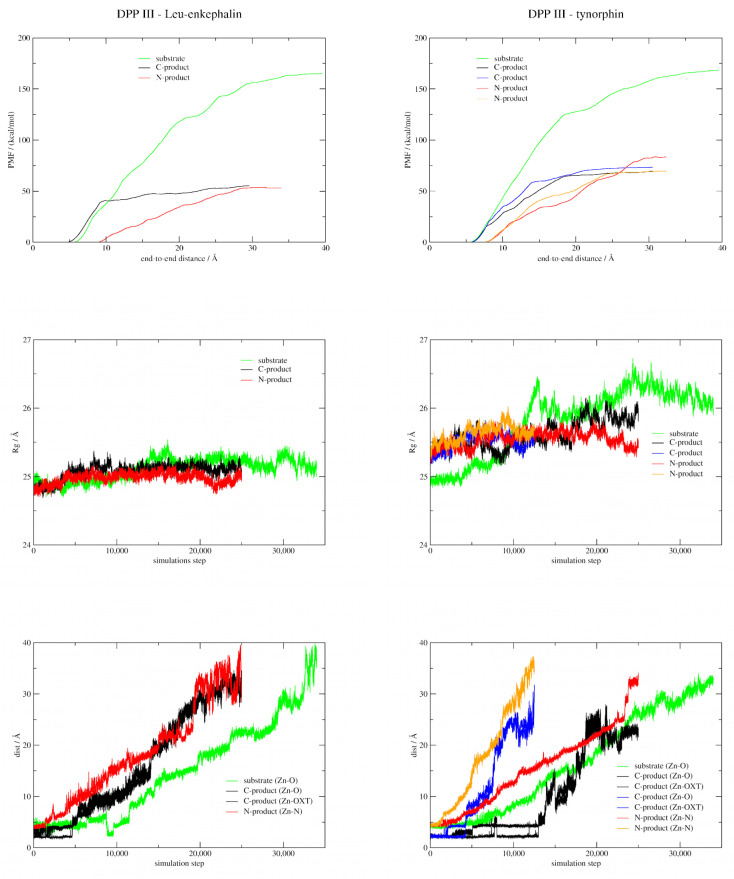
PMF profiles for the release of substrates and products from the enzyme binding site into the solvent environment obtained from ASMD simulations. The corresponding gyration radius profiles (Rg) and the distances of the carbonyl oxygen atom from the cleavable peptide bond of the substrate, the carboxylate oxygen atoms of the C-products and the amine nitrogen atoms of the N-products from the zinc ion are also shown. The reaction coordinate is the end-to-end distance between a) the heavy atoms of the backbone of ligand P2 and the β-strand of the lower domain of the protein (residues Ala388-Asn391) when substrates (green) and tynorphin C-product (black, *v* = 0.5 Å/ns, and blue, *v* = 1 Å/ns) are released from the enzyme binding site, b) the zinc ion and the center of mass of the backbone heavy atoms of the amino acids from the cleavable peptide bond to the N-terminus of the ligand (P1 and P2) in the case of release of the Leu-enkephalin C-product (black), and c) the zinc ion and the center of mass of the heavy atoms of the amino acids found from the scissile peptide bond to the C-terminus of the ligand (P1′, P2′ and P3′) in the case of N-product release (red, *v* = 0.5 Å/ns, and orange, *v* = 1 Å/ns).

**Table 1 ijms-23-01858-t001:** Selected bond lengths and dihedral angles during tynorphin peptide bond cleavage in the active site of human DPP III. Distances used as reaction coordinates are in bold. Calculations at the B97D/[6-31G(d) + LanL2DZ-ECP] level of theory. The names of the atoms correspond to those in Figure 1.

	ES	TS1	INT1	TS2	INT2	TS3	INT3	TS4	EP
*d*(O_s_-Zn)/Å	3.00	2.09	2.12	2.16	2.24	2.80	2.89	2.98	3.00
*d*(O_w_-Zn)/Å	1.96	2.15	2.12	2.10	2.16	2.01	2.00	1.99	1.99
*d*(C_s_-N_s_)/Å	1.35	1.45	1.53	1.58	2.56	2.91	3.08	3.08	3.16
*d*(O_w_-C_s_)/Å	2.59	1.55	1.47	1.45	1.29	1.29	1.28	1.28	1.29
***d*****(O**_**s**_**-C**_**s**_**)**/**Å**	**1.25**	**1.35**	**1.35**	**1.34**	**1.28**	**1.26**	**1.26**	**1.26**	**1.26**
*d*(O_w_-H_w2_)/Å	0.98	0.98	1.00	1.01	1.76	2.36	2.99	2.95	2.23
*d*(O_w_-H_w1_)/Å	1.33	2.14	2.58	2.46	2.62	3.87	4.23	4.22	3.97
*d*(H_w2_-O_e2_)/Å	2.32	2.27	1.75	1.63	1.01	1.00	1.04	1.40	2.97
*d*(H_w1_-O_e1_)/Å	1.13	1.00	1.05	1.19	2.05	2.12	2.44	2.19	1.06
***d*****(N**_**s**_**-H**_**w1**_**)**/**Å**	**3.03**	**2.19**	**1.71**	**1.39**	**1.02**	**1.02**	**1.02**	**1.03**	**1.58**
***d*****(N**_**s**_**-H**_**w2**_**)**/**Å**	**3.68**	**2.91**	**2.61**	**2.62**	**3.45**	**2.42**	**1.75**	**1.20**	**1.02**
*d*(H450-Zn) ^a^/Å	2.11	2.12	2.11	2.11	2.09	2.08	2.07	2.07	2.07
*d*(H455-Zn) ^a^/Å	2.12	2.12	2.13	2.12	2.12	2.10	2.09	2.09	2.09
*d*(E508-Zn) ^a^/Å	2.03	2.04	2.03	2.02	2.03	2.01	2.00	2.01	2.01
*d*(H568[Hε]-O_s_)/Å	1.69	1.60	1.64	1.67	1.70	1.65	1.67	1.68	1.65
*d*(H568[Nε-Hε])/Å	1.05	1.07	1.07	1.06	1.05	1.05	1.05	1.05	1.05
*d*(Y318-sub) ^b^/Å	1.90	1.90	1.91	1.91	1.98	1.00	2.04	2.05	2.03
*ω*_1_ (C-N_s_-C_s_-H_s_) ^c^/°	−172.6	151.7	128.2	124.9	-	-	-	-	-
*ω*_2_ (C′-C_s_-N_s_-O_s_) ^c^/°	170.0	132.1	130.2	129.6	-	-	-	-	-

^a^ Distances between the zinc ion and either nitrogen (Nδ) or oxygen (carboxyl) atoms from the histidine of glutamate amino acid residues, respectively. ^b^ Distance between the oxygen atom from the Tyr318 hydroxyl group and amide hydrogen atom from the second amino acid residue from the substrate (sub) N terminus. ^c^ C and C′ are carbon atoms adjacent to Ns or Cs atom, respectively, while Hs is a hydrogen atom bonded to Ns (only dihedrals values before peptide bond cleavage are shown).

**Table 2 ijms-23-01858-t002:** Enthalpy contribution to the binding free energies calculated using the MM/PBSA approach (the solute dielectric constant was 2 and that of solvent 80) for the last 40 ns of MD simulations.

Complex	Receptor	Ligand	Δ*H*/(kcal/mol)	SD/(kcal/mol)
DPP III–Leu-enkephalin	DPP III + WAT	Leu-enkephalin	−17.73	4.83
DPP III–tynorphin	DPP III + WAT	tynorphin	−26.56	5.33
DPP III–Leu-enkephalin product	DPP III + C-prod	N-prod	−6.97	3.80
DPP III–tynorphin product	DPP III + C-prod	N-prod	−27.88	2.93
DPP III–Leu-enkephalin product	DPP III + N-prod	C-prod	−18.13	4.56
DPP III–tynorphin product	DPP III + N-prod	C-prod	−34.40	3.43

## Data Availability

Not applicable.

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
