# Peer review of "Demystifying DPP III Catalyzed Peptide Hydrolysis—Computational Study of the Complete Catalytic Cycle of Human DPP III Catalyzed Tynorphin Hydrolysis"

_ijms, 2022, doi:10.3390/ijms23031858_

Round 1

Reviewer 1 Report

Dear Authors,

In the current report Tomic et al., have described entire catalytic cycles if human DPP III by using quantum molecular mechanics calculations and various molecular dynamics. Their analysis showed that catalytic process of DPP III follows same reaction with substrates, tynorphin and Leu-enkephalin. However, the reaction difference between the two products depends on higher stabilization of the products and thus, lower regeneration of enzyme. Article is very well written, and all the calculations are well performed by the authors.

There are few concerns which could be addressed:

  • One of the major concerns came across is to find out the study’s importance in terms of validation. Throughout study was performed via CANVER and AMBER software. Is there any wet lab mechanistic assay performed before or after?
  • Also, it is worth seeing other substrates study other than tynorphin and Leu-enkephalin.
  • What binding assays were performed to find the data? What was the reference binding range value given?
  • Is there any pharmacokinetics or pharmacodynamics data were analyzed as these are the important factors for drug design and development?
  • The compliance parameters could also be monitored.
  • Since tynorphin is poor DPP III substrate. Is there any modification, structure or ligand-based design assays were calculated to improve?
  • Computational study is up to 70% accurate. How could authors justify the accuracy of data?

My suggestion is Minor revision.

Reviewer 2 Report

Referee report concerning the manuscript

ijms-1554646

Demystifying DPP III catalysed peptide hydrolysis - computational study of the complete catalytic cycle of human DPP III catalysed tynorphin hydrolysis,

by A. Tomic and S. Tomic

The authors studied chemical step of the enzyme DPP III, a zinc dependent exopeptidase that hydrolyzes peptides of various sequence and size by using DFT QM/MM methodology. Decomposition pathways of two peptides were studied: Leu-enkephalin and tynorphin. Latter is a truncated model of endogenous heptapeptide spinorphin. Both reaction profiles were calculated by energy minimization along the postulated reaction coordinate for several reaction steps. It was demonstrated that for enkephalin decomposition the rate limiting step is formation of the tetrahedral complex when hydroxide ion attacks scissile peptide bond carbon atom. In the case of tynorphin decomposition formation of the hydroxide ion from a water molecule proved to be the rate limiting step. I would like to emphasize that presented calculations are extremely demanding since they involve heterolytic decomposition of a water molecule, what is an equivalent of an SN1 reaction and therefore agreement with the experimental barrier is not perfect. The study is relevant for understanding properties of the enzyme DPP III.

The manuscript represent an important new contribution and it should be published.

There are however few weak points.

1) I took pencil and paper and I calculated from the experimental kinetics of tynorphin decomposition on page2 (4 hours to be decomposed at 37C) the activation free energy value of 24.3 kcal/mol. I took half time of 4 hours. The authors report in the manuscript values of activation energy of 14.21 kcal/mol and Figure 2 shows activation energies of about 12 kcal/mol for both substrates.

Comment!

2) The barrier values of 14.21 kcal/mol and 12 kcal/mol should include formation of OH- that is very much pH dependent. There is analytical expression for free energy cost for formation of OH-

Formation of OH- then is heavily pH dependent and the kinetics should be in turn very much pH dependent,

Is something known experimentally about the pH dependent rate constant for both substrates?

At room temperature and pH value of 7.4 formation of

OH- in bulk water costs k_B*T ln(10)* (15.7 − pH) = 11.3 kcal/mol, where 15.7 is water pKa value. As a simple example of the pH dependent reactions involving OH- see:  Mol. NeuroSci. 11 (2018) article 467.

It would be a major challenge to calculate pKa value of reactive water in the active site in the presence of this or that substrate.

I would add few sentences that pKa calculations are extremely demanding. Several readers do not understand that this is formally an SN1 process and pKa calculations represent one of the most demanding tests for applied molecular simulation protocol.

  1. Phys. Chem. B 1997, 101, 4458-4472

Quote, comment!

3) Figure 2 shows that in the case of enkephalin formation of the tetrahedral complex is rate-limiting, while in the case tynorphin formation of OH- is rate limiting. I would not be sure that change of the substrate changes pKa of a water molecule at the active site that much. See also my comment #2.

Comment!

4) A challenge for future is calculation of the rate limiting step on the level of  Empirical Valence Bond, that allows for thermal averaging and provides well-converged results. Ab initio QM/MM currently does not provide well-converged results.

  1. Warshel, Computer Modelling of Chemical Reactions in Enzymes

and Solutions, John Wiley and Sons, 1991, New York

The authors may think about using Empirical Valence Bond methodology as implemented in Q6.

The SW is available free of charge.

Bauer, P., Barrozo, A., Purg, M., Amrein, B.A., Eguerra, M., Barrie Wilson, P., Major, D.T., Åqvist, J., Kamerlin, S.C.L. (2018) SoftwareX DOI: 10.1016/j.softx.2017.12.001 "Q6: Acomprehensive toolkit for empirical valence bond and related free energy calculations".

https://www.icm.uu.se/cbbi/aqvist-lab/q/

Quote, comment!

5) I would shorten the manuscript.

--End of comments--
